# A Modified Johnson-Cook Model for Ferritic-Pearlitic Steel in Dynamic Strain Aging Regime

**Ashwin Moris Devotta [1,2,\*], P. V. Sivaprasad [3], Tomas Beno [2], Mahdi Eynian [2] , Kjell Hjertig [2] , Martin Magnevall [4,5] and Mikael Lundblad [4]**

[1] R&D Turning, Sandvik Coromant AB, 811 81 Sandviken, Sweden
[2] Department of Engineering Science, University West, SE-461 32 Trollhättan, Sweden;
   tomas.beno@hv.se (T.B.); mahdi.eynian@hv.se (M.E.); kjell.hurtig@hv.se (K.H.)
[3] R&D, Sandvik Materials Technology AB, 811 81 Sandviken, Sweden; palla.sivaprasad@sandvik.com
[4] R&D, Sandvik Coromant AB, 811 81 Sandviken, Sweden; martin.magnevall@sandvik.com or
   martin.magnevall@bth.se (M.M.); mikael.lundblad@sandvik.com (M.L.)
[5] Department of Mechanical Engineering, Blekinge Institute of Technology, SE-371 41 Karlskrona, Sweden
[\*] Correspondence: ashwin.devotta@sandvik.com; Tel.: +46-706-163-722

**Abstract:** In this study, the flow stress behavior of ferritic-pearlitic steel (C45E steel) is investigated through isothermal compression testing at different strain rates (1 s$^{-1}$, 5 s$^{-1}$, and 60 s$^{-1}$) and temperatures ranging from 200 to 700 °C. The stress-strain curves obtained from experimental testing were post-processed to obtain true stress-true plastic strain curves. To fit the experimental data to well-known material models, Johnson-Cook (J-C) model was investigated and found to have a poor fit. Analysis of the flow stress as a function of temperature and strain rate showed that among other deformation mechanisms dynamic strain aging mechanism was active between the temperature range 200 and 400 °C for varying strain rates and J-C model is unable to capture this phenomenon. This lead to the need to modify the J-C model for the material under investigation. Therefore, the original J-C model parameters *A*, *B* and *n* are modified using the polynomial equation to capture its dependence on temperature and strain rate. The results show the ability of the modified J-C model to describe the flow behavior satisfactorily while dynamic strain aging was operative.

**Keywords:** flow stress; modified Johnson-Cook model; dynamic strain aging

## 1. Introduction

Both from an economic and engineering view point, modeling of the machining process is important. Within modeling of the machining process, numerical approaches such as finite elements (FE) have gained prominence in the past decades. FE modeling of machining process has matured to the level of predicting industrially relevant output parameters (e.g., cutting forces, tool wear, residual stresses). During machining, the workpiece material undergoes large strains (~5) at high strain rates (~10$^5$ s$^{-1}$) and temperatures (~800 °C) [1]. C45E steel which belongs to the ferritic-pearlitic steel group is one of the most commonly machined steels in general engineering applications. It has also been used in the development and evaluation of FE models that predict chip formation in machining. Various constitutive models have been reported in the literature that can be used with these FE models. Melkote et al. [2] have provided an overview of the different constitutive models used in modeling of the machining process. Johnson-Cook (J-C) model [3] is one of the most commonly used phenomenological models where flow stress is described as a function of strain, strain rate and temperature. J-C model has been used to study the high strain rate material behavior of different materials [4–7]. In the J-C model, the demarcation of influencing factors (strain, strain rate, and temperature) gives an advantage from a

numerical modeling viewpoint. At the same time, this also leads to missing the interactions among influencing factors from an engineering viewpoint. Jaspers and Dautzenberg [8] conducted high strain rate deformation tests at varying temperatures for AISI 1045 steel and modeled the flow stress using J-C and Zerilli-Armstrong models. The J-C parameters so obtained have been used in several reported works in the modeling of chip formation in machining. However, they specify that Zerilli-Armstrong is better in the fitting of flow stress curves for AISI 1045 steel. Recently, Abouridouane et al. [9] have conducted similar material tests and used the J-C model to fit the flow stress curves in C45E steel to study chip formation during micromachining.

From a metallurgical perspective, in ferritic-pearlitic steels, different deformation mechanisms such as strain hardening, dynamic recovery, dynamic recrystallization, and dynamic strain aging (DSA) are active at certain temperature—strain rate ranges. Therefore, modeling the deformation behavior of the ferritic-pearlitic steels should be able to capture the different deformation mechanisms. Courbon et al. [10] conducted compression tests at varying temperatures in C45E steel using Gleeble and showed that J-C model is unable to predict the softening behavior observed in the primary shear zone during adiabatic shear band formation. To overcome the original J-C model's shortcomings, the J-C model of C45E steel has been modified for specific strain rate-temperature ranges [11–15]. Sartkulnavich et al. [11] modified the temperature component of J-C model to include the blue brittleness between the temperature range 500 and 700 °C but assumed no influence of strain rate on blue brittleness. Li et al. [12] replaced the J-C model's temperature component with an exponential function to better model the temperature influence on flow stress. To predict residual stresses in machining, Ee et al. [13] modified the strain rate component to include the influence of both low and high strain rate deformation mechanism. Hor et al. [16] modified the strain hardening component of ferritic-pearlitic steel's J-C model using a damage model to predict chip segmentation. However, these modifications have not been able to capture the influence of the presence of DSA phenomenon of C45E steel. During DSA, it has been reported that the flow stress increases with temperature and decreases with strain rate in contrary to the normal expected behavior and has been observed in HSLA-65 steel with a ferrite-pearlite microstructure [17]. The other signatures of DSA are serrations in the flow curve etc. Kim and Kang [18] have observed the serrations in flow stress curves at very low strain rates in SA508-class 3 pressure vessel steel with ferrite-pearlite microstructure.

In this work, an attempt is made to model the C45E steel's flow stress behavior by modifying the J-C model where the DSA phenomenon occurs. To the best of our knowledge, this is first ever attempt to describe the flow behavior of this steel using a phenomenological model (J-C model) under the influence of DSA. To consider the anomalous flow behavior of C45E steel, the J-C model is suitably modified and discussed in this paper.

## 2. Experimental Method

The chemical composition of C45E steel used in this investigation is given in Table 1. It was obtained as rods with a diameter of 20 mm and an initial hardness of 202 HB. The rods were machined within Gleeble specified geometrical tolerances into compression specimens with diameters of 4 mm and 10 mm with an l/*d* ratio of 1.2. Hot compression tests were carried out using the Gleeble thermomechanical simulator, Gleeble 3800-GTC, (Dynamic Systems Inc., Postenkill, NY, USA) where cylindrical specimens were deformed at programmed strain rates of $1\,s^{-1}$, $5\,s^{-1}$, and $60\,s^{-1}$. A schematic of the Gleeble machine used for the compression is shown in Figure 1a. 4 mm diameter specimens were used for low-temperature range (200 °C, 300 °C, 400 °C) and 10 mm diameter specimens were used for the remaining test conditions of 500 °C, 600 °C, and 700 °C considering the machine's capacity. However, owing to the machine capacity the tests conducted at 200 °C, 300 °C, and 400 °C could not be performed until 50% height reduction. Thermocouples were welded at the surface center of the specimen as shown in Figure 1a. The Gleeble thermomechanical simulator provides the possibility to heat the samples directly between the anvils using direct resistance heating. A schematic (Figure 1b) showing the heating cycle and the compression is provided. The specimen heating rate was kept constant

at 5 °C·s⁻¹. The specimens were held at the programmed temperature for 180 s to ensure thermal equilibrium. Since the temperatures at which the tests were conducted was $0.5 \times T_m$ ($T_m$—melting temperature) of the C45E steel the heating rate was not expected to lead to microstructural changes [19]. The test was repeated twice and in case of considerable variation was observed, the test was repeated once more. The experimental results with maximum repeatability were chosen for further processing.

**Table 1.** Major alloy content (wt. %) of C45E steel used in the study.

| C | Si | Mn | P | S | Cr | Ni | Mo | Fe |
|------|------|------|-------|-------|------|------|------|---------|
| 0.44 | 0.24 | 0.66 | 0.006 | 0.031 | 0.13 | 0.19 | 0.03 | Balance |

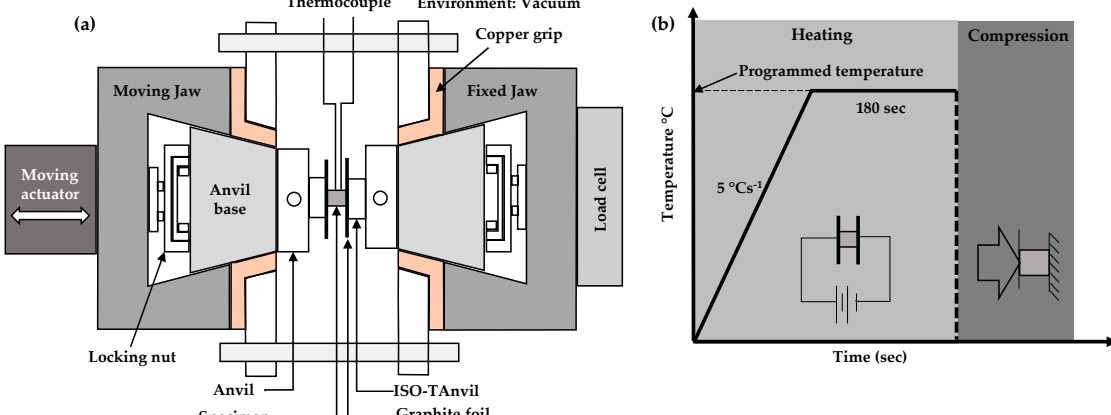

**Figure 1.** (**a**) Schematic of the Gleeble hot compression testing setup with thermocouples welded to the specimen; (**b**) heating and compression cycle used in the study.

The tests were conducted at a constant strain rate mode for the programmed strain rates of 1 s⁻¹ and 5 s⁻¹ and a constant velocity mode for the programmed strain rate of 60 s⁻¹. At lower programmed strain rates of 1 s⁻¹ and 5 s⁻¹, Gleeble machine's controller can adjust the varying displacement requirement to maintain a constant strain rate (constant strain rate mode). At the higher strain rate of 60 s⁻¹, the deformation time is short for the machine's closed-loop control system to vary the displacement and hence a constant velocity is set (constant velocity mode). In addition to the compression tests at elevated temperatures, one compression test was carried out at room temperature at a strain rate of 1 s⁻¹. The testing was carried out under vacuum. The machine's output in the form of strain, stress, load, and displacement with a time stamp was obtained.

Gleeble post-processing software calculates the true stress and true strain. This post-processing includes both elastic and plastic deformation. To remove the elastic strain component and obtain the true stress versus plastic strain curve, the elastic portion of the load-displacement curve was deducted. The resulting load-displacement curve was used for further processing. The true stress ($\sigma$) was calculated from the resulting load-displacement curve using the formula as shown in Equation (1):

$$\sigma = \frac{P}{A} = \frac{4P}{\pi D^2} = \frac{4Ph}{\pi D_0^2 h_0} \tag{1}$$

where $P$ is the machine applied load, $A$ is the instantaneous cross-sectional area, $h$ is the specimen height being measured continuously by the machine's axis position and using a linear variable differential transformer (LVDT) type hot zone transducer, $h_0$ is the initial specimen height, $D$ is the instantaneous specimen diameter and $D_0$ is the initial specimen diameter. True plastic strain ($\varepsilon$) is calculated as shown in Equation (2):

$$\varepsilon = \ln \frac{h_0}{h} \tag{2}$$

The obtained true stress versus true plastic strain for varying temperature and strain rate was used to model the flow stress behavior of C45E steel.

## 3. Results and Discussion

### 3.1. Experimental Results

Compression tests are conducted for C45E steel specimens and flow stress curves in the form of true stress versus true plastic strain are obtained. Figure 2 shows the true stress-true plastic strain at room temperature at a strain rate of 1 s$^{-1}$. Due to the limitation of the machine's load capacity, the test was limited to a strain of 0.08 where the flow stress curves showed continuous strain hardening. The room temperature true stress–true plastic strain is used later in the study for the fitting of the original J-C model.

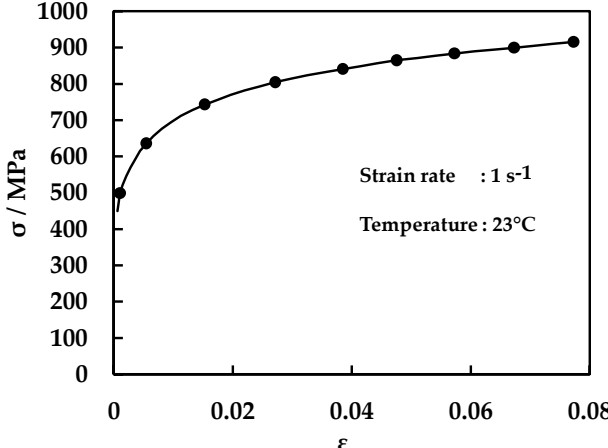

**Figure 2.** True stress versus true plastic strain at a strain rate of 1 s$^{-1}$ at room temperature.

The flow stress curves at programmed temperatures between 200 °C and 700 °C and programmed strain rates of 1 s$^{-1}$, 5 s$^{-1}$, and 60 s$^{-1}$ is presented in Figure 3a–c. Two temperature zones can be seen in Figure 3. One zone is between 200 °C and 400 °C where an anomalous behavior is seen concerning the variation of flow stress with temperature and strain rate. The other zone is between 500 °C and 700 °C where the flow behavior is as expected with temperature (thermal softening). The strain hardening behavior also varies in the two temperature zones of 200–400 °C and 500–700 °C. In the temperature zone of 200–400 °C, the material strain hardens continuously leading to a continuous increase of flow stress with an increase in strain. However, in the temperature zone, 500–700 °C, the flow stress shows stress saturation showing recovery mechanisms are operative. This means in the temperature zone of 500–700 °C, strain hardening competing with softening by dynamic recovery leads to constant flow stress with an increase in strain. With the strain rate increase from 1 s$^{-1}$ to 5 s$^{-1}$ and 60 s$^{-1}$ in the programmed temperature zone of 200 °C to 400 °C, the reduced flow stress indicates negative strain rate sensitivity. For better visualization of the combined effect of temperature and strain rate sensitivity, the flow stress at a strain of 0.1 is used in the following section.

Figure 4 shows the flow stress variation at a strain of 0.1 for varying temperatures and strain rates. The flow stress at a strain of 0.1 is chosen to avoid the transient conditions present during the early stages of loading. It is in order to mention that serrated flow was observed in some of the stress-strain curves. To obtain the true stress value at a plastic strain of 0.1 consistently, the experimental data points are fitted using a 6th order polynomial with strain as a variable. This procedure is carried out to remove the undulations due to serrated flow. This is the reason for the minor variations in the true stress at a strain of 0.1 ($\sigma_{0.1}$) in Figures 4, 6, and 7 from the true stress at a strain of 0.1 ($\sigma_{0.1}$) in Figure 3. Figure 4 shows the flow stress variation in the two temperature zones of 200–400 °C and 500–700 °C. The flow stress increases in the temperature zone 200–400 °C and decreases with further temperature

increase in the temperature zone 500–700 °C. At strain rates 1 s$^{-1}$ and 5 s$^{-1}$, the flow stress increases in the temperature zone 200–400 °C. On the other hand, at the strain rate of 60 s$^{-1}$, the flow stress decreases marginally from 200 °C to 300 °C and slightly increases from 300 °C to 400 °C.

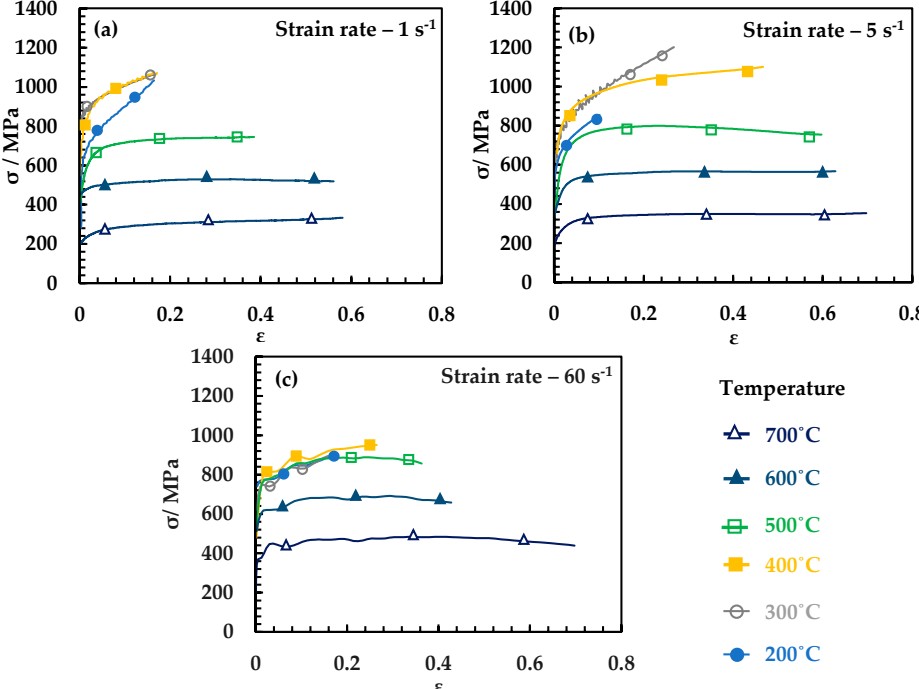

**Figure 3.** Flow stress at strain rate 1 s$^{-1}$ (**a**), 5 s$^{-1}$ (**b**), and 60 s$^{-1}$ (**c**) with the temperature between 200 °C and 700 °C.

The strain rate's influence is also varied in the temperature zone 200–400 °C and 500–700 °C. Negative strain rate sensitivity is observed in the 200–400 °C temperature zone whereas positive strain rate sensitivity is observed in the 500–700 °C temperature zone.

The results may be summarized as follows:

(a)   Two temperature zones were identified with different flow behavior where the flow stress increased with temperature in the temperature zone (200–400 °C) and the flow stress decreased with temperature in the temperature zone (500–700 °C).

(b)   The flow stress decreased with increase in strain rate in the temperature range 200 °C to 400 °C.

(c)   The serrated flow was observed in the temperature range 200 °C to 300 °C.

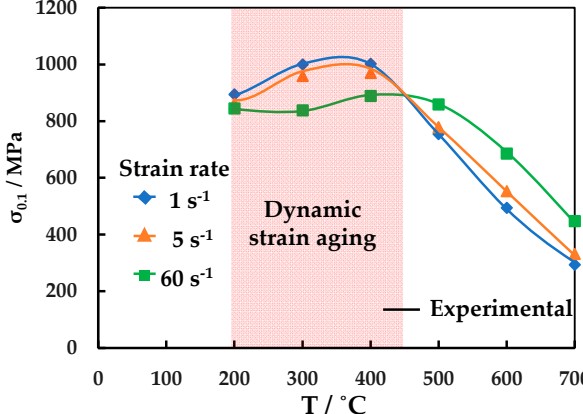

**Figure 4.** Flow stress at a true plastic strain of 0.1 versus temperature.

### 3.2. Fitting of Flow Stress Curves to Original J-C Model

As part of this work, the flow stress curves obtained through the experimental investigation is fitted to the standard J-C model (Equation (3)), where *A*, *B*, *n*, *C*, and *m* are J-C parameters to be fitted. Using internally developed genetic algorithm-based optimization routine, values of the J-C model parameters are obtained and is shown in Table 2.

$$\sigma = (A + B\varepsilon^n)\left(1 + C\ln\left(\frac{\dot{\varepsilon}}{\dot{\varepsilon}_0}\right)\right)\left(1 - \left(\frac{T - T_0}{T_m - T_0}\right)^m\right) \tag{3}$$

**Table 2.** J-C parameters fitted using Genetic algorithm (GA) approach and Jaspers and Dautzenberg reported values.

| J-C Variant | A | B | n | C | m |
|---|---|---|---|---|---|
| GA Approach | 544 | 332 | 0.158 | 0.010 | 1 |
| Jaspers and Dautzenberg | 553 | 600 | 0.234 | 0.013 | 1 |

In this approach, each component of the J-C model is not separately calibrated whereas in the well-established methodology for the fitting of the J-C model each of its component is separately calibrated. The values obtained in Table 2 are comparable to the values obtained in [8] used extensively in the modeling of chip formation in the machining process. This confirms the accuracy of the internally developed genetic algorithm-based optimization routine used to predict the J-C model parameters. Figure 5 shows the flow stress curves predicted using the original J-C model parameters in Table 2 along with the experimentally obtained flow stress curves at a strain rate of $1\ \text{s}^{-1}$. The strain, strain rate and temperature values measured during the compression tests are used to calculate the corresponding stress values using the original J-C model. Hence the strain rate values and temperature values are not constant in the graphs provided in Figure 5. Figure 5 shows that the original J-C model fits poorly the experimental flow stress curves.

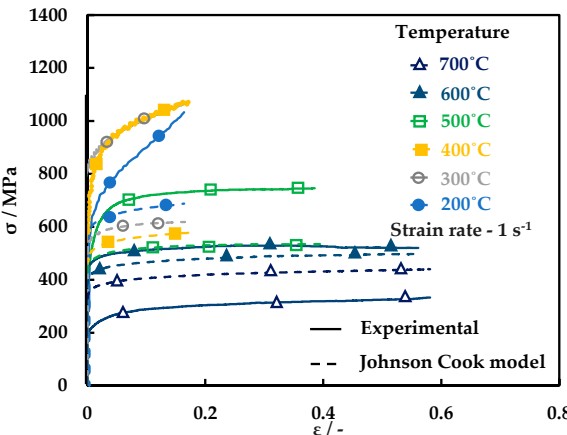

**Figure 5.** Flow stress curves fitted with J-C model.

The original J-C model is unable to capture the flow stress for varying temperatures and strain rates. The results also show that the original J-C model is not able to predict the negative strain rate sensitivity and negative thermal softening in the temperature zone 200–400 °C for the strain rates $1\ \text{s}^{-1}$, $5\ \text{s}^{-1}$, and $60\ \text{s}^{-1}$.

### 3.3. Influence of Temperature and Strain Rate on Flow Stress at a Strain of 0.1

With the fitting of experimental results with the original J-C model showing poor fit, the flow stress at varying temperature and strain rate is analyzed. Figure 6 compares between the J-C model

predicted flow stress with experimentally obtained flow stress at a strain of 0.1. It can be clearly seen that the J-C model cannot predict the increase in flow stress with an increase in temperature zone 200–400 °C. Similarly, it cannot predict the negative strain rate sensitivity in the above temperature zone. The flow stress increase in the temperature zone 200–400 °C and the negative strain rate sensitivity has been observed as an anomalous behaviour in the experimental results. This anomalous behaviour is attributed to DSA phenomenon. Literature research shows that C45E steel exhibits DSA (Hor et al. [20] and Hokka et al. [21]) at certain strain rate—temperature ranges. The flow stress analysis at a strain of 0.1 (Figure 6) shows the presence of dynamic strain aging in the tested temperature range of 200–400 °C and strain rate range of 1 s$^{-1}$, 5 s$^{-1}$, and 60 s$^{-1}$. When different slip systems interact, they form dislocation junctions. These dislocation junctions are prime regions for diffusion and accumulation of interstitial solute atoms such as carbon and nitrogen which leads to trapping of mobile dislocations making them immobile. This dislocation motion resistance is partly responsible for DSA in steels resulting in the macroscopic behavior such as increased flow stress with an increase in temperature, negative strain rate hardening and serrations in the stress-strain curve. With DSA having shown to influence the fracture behavior of steels (Li and Leslie [22]), its need to be incorporated in the flow stress behavior of steels is clear. But, the reported works where C45E steel has been modeled by the J-C model have not considered the presence of DSA [23] and modified J-C models have not taken it into account either. At 1 s$^{-1}$ and 5 s$^{-1}$, the continuous flow stress increase in the temperature zone (200–400 °C) and the observed serrations show a strong presence of DSA. Whereas at 60 s$^{-1}$, the absence of serrated flow and a marginal flow stress increase from 300–400 °C shows a relatively weaker presence of DSA. This clearly shows that the DSA regime is strongly influenced by the strain rate-temperature range. To capture DSA active in the above strain rate-temperature range, the J-C parameters *A* and *B* are to be modeled as a function of temperature and strain rate.

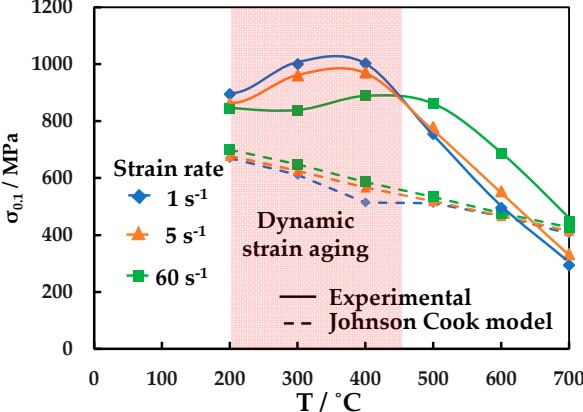

**Figure 6.** Flow stress at a true plastic strain of 0.1 versus temperature modeled using the original J-C model.

### 3.4. Fitting of Flow Stress Curves to Modified J-C Model

To incorporate the influence of DSA and varying strain hardening behavior, the original J-C model needs to be modified. In J-C model, parameter *A* is responsible for the initial yield stress while parameters *B* and *n* are responsible for strain hardening behavior. The original J-C model's parameters *A* and *B* being constant is unable to predict the influence of DSA regime. The following section discusses the modification of parameters *A* and *B* to consider the flow stress variations due to DSA.

Qingdong et al. [24] used linear polynomial functions of homologous temperature ($T^*$) as shown in Equation (4) to model the original J-C model's parameters (*A* and *n*).

$$\sigma = A(1 + pT^*) + B\varepsilon^{n_0 + n_1 T^*} \tag{4}$$

From Figure 6, we can see that the negative thermal softening in the temperature range (200–400 °C) and positive thermal softening in the temperature range (500–700 °C) cannot be captured with a power function of Qingdong et al. [24].

In this work, the flow stress curve for each combination of strain rate and temperature is fitted with the J-C model's strain hardening component ($A + B\varepsilon^n$). The initial yield stress value, "$A$" is obtained for each test condition. To fit the parameters, $B$ and $n$, the experimental true stress versus plastic strain curves are fitted as $\ln(\sigma - A_i) = B + \ln\varepsilon$. This provides the strain hardening coefficient, $B$ and strain hardening exponent, $n$, for each test condition. This leads to 18 sets of J-C model strain hardening constants. Subsequently, the parameters, $A$ and $B$ are fitted with a second order polynomial function with homologous temperature and $\ln\left(\frac{\dot{\varepsilon}}{\dot{\varepsilon}_0}\right)$ as shown in Equations (5) and (6) respectively. Least square method is used to fit the constants, $A_0$–$A_4$ and $B_0$–$B_4$ and is presented in Table 3. This enables to capture the trend shown in Figure 6.

$$A = A_0 + A_1 \ln\left(\frac{\dot{\varepsilon}}{\dot{\varepsilon}_0}\right) + A_2 T^* + A_3\left[\ln\left(\frac{\dot{\varepsilon}}{\dot{\varepsilon}_0}\right)\right]^2 + A_4 T^{*2} \tag{5}$$

$$B = B_0 + B_1 \ln\left(\frac{\dot{\varepsilon}}{\dot{\varepsilon}_0}\right) + B_2 T^* + B_3\left[\ln\left(\frac{\dot{\varepsilon}}{\dot{\varepsilon}_0}\right)\right]^2 + B_4 T^{*2} \tag{6}$$

**Table 3.** Parameter $A$ as a 2nd degree polynomial function of temperature and strain rate.

| $A_0$ | $A_1$ | $A_2$ | $A_3$ | $A_4$ | $B_0$ | $B_1$ | $B_2$ | $B_3$ | $B_4$ | $n_0$ | $n_1$ |
|---|---|---|---|---|---|---|---|---|---|---|---|
| −763.44 | 41.04 | 9691 | −2.46 | −14.674 | 1390.6 | 182.6 | −3060.7 | −59.27 | 1751.24 | 0.07 | 1 |

In this study, the modified J-C model is fitted for validity between temperatures 200 °C and 700 °C. This results in $A_0$ being negative whereas in the original J-C model the value of $A$ corresponds to yield stress at the reference strain rate and room temperature. This leads to the value of $A$ in the original J-C model to be always positive whereas in the modified J-C model, the value of $A_0$ does not have any physical significance. Similarly, the parameter $B$ is also defined as a second-order polynomial function with homologous temperature and strain rate as parameters as shown in Equation (6). Like Qingdong et al., the strain hardening exponent, $n_i$ is defined as a first-degree polynomial function of homologous temperature ($T^*$) in this work with the constants, $n_0 = 0.07$ and $n_1 = 1$ using least squares method. Figure 7 shows the modified J-C model predicted flow stress at a strain of 0.1. It shows the modified parameters ($A$, $B$, and $n$) captures the trend in flow stress variation due to DSA compared to the original J-C model (Figure 6). The flow stress increase with a temperature increase and strain rate decrease in the DSA regime is captured qualitatively. The flow stress at 300 °C and 400 °C is higher than that at 200 °C. However, there are differences in the absolute values for the strain rate of 1 s$^{-1}$. The modified J_C model is shown in Equation (7) with the modified strain hardening component. The strain rate hardening parameter '$C$' and thermal softening parameter '$m$' are fitted using the approach for the original J-C model.

$$\sigma = \left(A_i + B_i\varepsilon^{n_0 + n_1 T^*}\right)\left(1 + C \ln\left(\frac{\dot{\varepsilon}}{\dot{\varepsilon}_0}\right)\right)(1 + T^{*m}) \tag{7}$$

Figure 8a shows the modified J-C model predicted true stress versus true plastic strain curve along with experimentally obtained true stress versus true plastic strain curves at a strain rate of 60 s$^{-1}$. The result shows that the modified J-C model predicts increased flow stress with an increase in the temperature from 200 °C to 400 °C due to DSA. The modified J-C model is also able to predict the varying strain hardening behavior in the two temperature ranges. Figure 8b shows the negative strain rate sensitivity being captured at 200 °C. The negative strain rate sensitivity leads to the flow stress decrease with an increase in strain rate from 5 s$^{-1}$ to 60 s$^{-1}$.

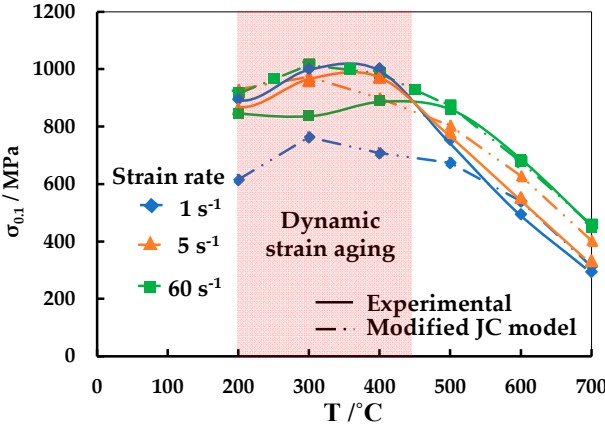

**Figure 7.** Flow stress at a true plastic strain of 0.1 versus temperature modeled using the modified J-C model.

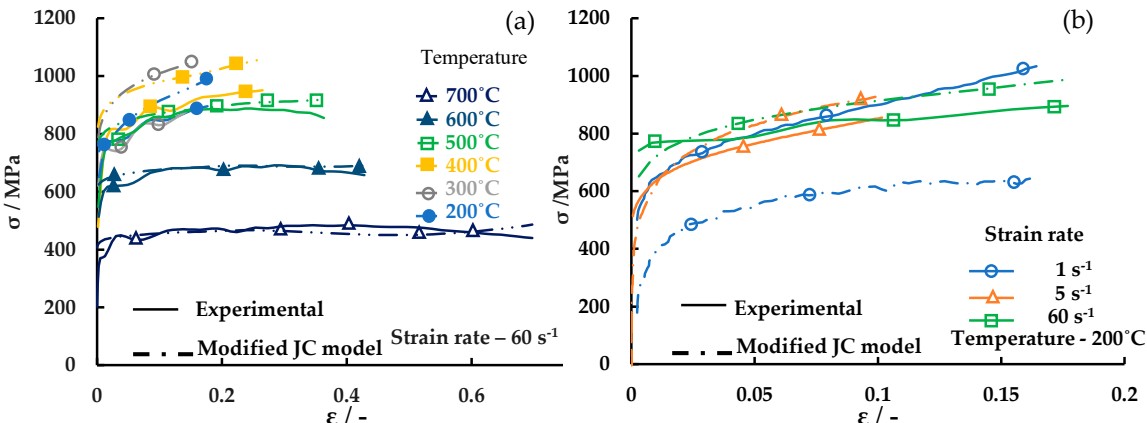

**Figure 8.** Modified J-C model predicted true stress versus true plastic strain curves (**a**) at different temperatures at a constant strain rate of 60 s$^{-1}$ and (**b**) at different strain rates at a constant temperature of 200 °C.

## 4. Conclusions

In this study, compression testing of C45E steel is carried out at temperatures between 200 °C and 700 °C and strain rates 1 s$^{-1}$, 5 s$^{-1}$ and 60 s$^{-1}$ using the Gleeble 3800 thermo-mechanical simulator and analyzed using true stress versus true plastic strain curves. The presence of varying deformation mechanisms is analyzed in terms of strain hardening, dynamic recovery, and DSA. The ability of the original J-C model to fit the experimentally obtained flow stress curve is tested and the strain hardening component is modified and a new modified J-C model is developed. The following conclusions are made from the study.

1. True stress versus true plastic strain of C45E Steel shows the strain hardening behaviour is dependent on the temperature with the presence of two temperature zones (200–400 °C and 500–700 °C) with strain hardening being higher in the temperature range 200–400 °C and dynamic recovery active in the temperature range 500–700 °C.
2. In the temperature range 200–400 °C, the presence of negative thermal softening, negative strain rate hardening and serrated flow indicates the presence of DSA.
3. The general J-C model is unable to capture the strain hardening variation and the presence of DSA.
4. The modification of J-C model's strain hardening component with the parameters *A* and *B* as a function of temperature and strain rate using a polynomial function enables capturing the DSA qualitatively.

**Author Contributions:** Conceptualization, A.M.D., P.V.S. and M.L.; methodology, A.M.D., P.V.S. and K.H.; validation, A.M.D., K.H. and M.L.; formal analysis, A.M.D., K.H. and P.V.S.; investigation, K.H. and A.M.D.; resources, M.M., T.B. and M.L.; data curation, A.M.D. and P.V.S.; writing—original draft preparation, A.M.D. and P.V.S.; writing—review and editing, P.V.S., M.E. and T.B.; visualization, A.M.D. and P.V.S.; supervision, T.B. and M.E.; project administration, A.M.D. and M.M.; funding acquisition, M.M. and M.L.

**Funding:** This research was funded by Sandvik Coromant AB and the Knowledge Foundation through the Industrial Research School SiCoMaP, Dnr 20110263, 20140130.

**Acknowledgments:** We would like to express our gratitude for the assistance provided by Simon Lövquist and Mikael Ekeroth, R&D, Sandvik Coromant, Sandviken, Sweden for the meticulous sample preparation used in the study.

**Conflicts of Interest:** The authors declare no conflict of interest.

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
