# Peer review of "A Modified Johnson-Cook Model for Ferritic-Pearlitic Steel in Dynamic Strain Aging Regime"

_metals, doi:10.3390/met9050528_

Round 1
Reviewer 1 Report
Review for metals-484756
A modified Johnson-Cook Model for Ferritic Pearlitic Steel in Dynamic Strain Aging Regime
The authors address a review dealing with an interesting topic for the journal. However, some recommendations should be considered:
Comments and Suggestions for Authors
[1] A better discussion on the effect of strain rate is recommended to clarify the explanation of Figures 3a, 3b and 3c.
[2] Figures 4, 6 and 7 consider the true plastic strain of 0.1, but the values represented in such figures are not consistent with those represented in Figure 3. Please, clarify this aspect.
Minor changes
[3] Line 30: “finite element (FE)” => finite elements (FE)
[4] Please, define LVDT in the main text (line 116).
Author Response
Dear Reviewer 1,
We would like to thank the reviewer for constructive evaluation and valuable comments. It helped greatly in revising the manuscript. Our response to the reviewer’s comments is given below.
The line numbers specified in the reply are applicable with the track changes mode ‘ON’ of the manuscript.
Improvement in methods, results and conclusion:
We have taken the reviewer’s input and guidance to improve the methods and Results section. We have rewritten the methods section on pages 7 & 8 for better clarity and have given more detailed information in modified JC model parameters identification. Similarly, the results section on pages 9 & 10 has also been improved to be clearer. The influence of strain rate on the flow stress curves has also been presented using Figure 3 (page 5: lines 164 -167). In the conclusion section, the limitation of the work is also added (page 11: line - 346).
Discussion on the effect of strain rate (Figures 3a, 3b and 3c)
Figure 3 contains the true stress vs plastic strain curves for 3 different strain rates and 6 temperatures. The strain rate’s influence on flow stress curves is better visualized by studying the true stress at a plastic strain of 0.1 and is used in Figure 4. In addition, in line with the reviewer’s comment, the influence of strain rate is given on page 5: lines 164 -167.
Calculation of True stress at a true plastic strain of 0.1
Figure 4 shows the variation of flow stress at a strain of 0.1 for varying temperatures and strain rates. It is in order to mention that serrated flow was observed in some of the stress-strain curves. To obtain the true stress value at a plastic strain of 0.1 consistently, the experimental data points are fitted using a 6th order polynomial with strain as a variable. This procedure is carried out to remove the undulations due to serrated flow. This is the reason for the minor variations in the true stress at a strain of 0.1 (σ0.1) in Figures 4, 6 and 7 from the true stress at a strain of 0.1 (σ0.1) in Figure 3.
The above text is added to the manuscript in Page 5: lines 168 – 173.
Other changes
(1) Line 30: “finite element (FE)” => finite elements (FE)
The suggested change is included in the manuscript. Page 1: line number - 31
(2) Please, define LVDT in the main text (line 116).
The suggested change is included in the manuscript. Page 4: line number - 134
We would like to thank the reviewer again for a very constructive review and suggestions for reviewing the work.
Thanking you very much in advance.
Yours sincerely
Ashwin Devotta
Sandviken, Sweden

Reviewer 2 Report
The paper appears to be well conceived.
More detailed description of the methods is needed
In some figures (plots) the fonts of the legend are a bit too small.
The reviewer would like to request moderate English revision. Please, consider to use a simpler language form to make the paper easier to understand to a wider audience.
Author Response
Dear Reviewer 2,
We would like to thank the reviewer for constructive evaluation and valuable comments. It helped greatly in revising the manuscript. Our response to the reviewer’s comments is given below.
The line numbers specified in the reply are applicable with the track changes mode ‘ON’ of the manuscript.
Improvement in Introduction section:
We have taken in the input from the reviewer and have changed significantly the introduction part in Page 1: lines 29 – 36,38 and Page 2: lines 46-47, 49-50,68-73. Readability is improved by changing the writing style to simple sentences.
Improvement in experimental, methods and results:
We have taken in the input from the reviewers in the need for improvement in Experimental, Methods and Results section.
In the experimental part, a schematic (Figure 1) (Page 3: line 110) is added instead of the photo and the schematic of the compression test procedure is added.
We have rewritten the methods section (page 9: line 230-237,255-260,273-284,292-295) for better clarity and have provided more detailed information. Similarly, the results section (page 10: line 297-304, page 10: line 309-310) has also been improved to be clearer.
Font legends:
In accordance with the comment, in the figures’ legend, BOLD fonts with increased font size have been used for better readability and to improve visualization.
English revision:
We have improved extensively the English changing complex sentences into simple sentences and reducing the percentage of passive voice. We also believe that further reducing the complex sentence could affect the readability (Page 1,2,3).
We would like to thank the reviewer again for a very constructive review and suggestions for reviewing the work.
Thanking you very much in advance.
Yours sincerely
Ashwin Devotta
Sandviken, Sweden

Reviewer 3 Report
This paper presents an investigation on the compressive behaviour of C45E steel at 3 strain rates and 6 temperatures.
Various comments will be useful for the authors:
1. Page 1. Abstract. Line 15. Instead of "experimental testing is post-", better is "experimental testing were post-".
2. Page 2. Line 84. Instead of "Diameter 4 specimens", better is "4 mm diameter specimens".
3. Page 2. Line 92. "heating rate was not expected to lead to microstructural changes". For so high temperatures (700 ºC), I dont think so.
4. Page 3. Figure 1. The Figure is not well understandable. It is important to also have one schematic drawing.
5. Page 3. Line 109. "calculate the machine’s compliance". How was the compliance (or stifness) measured? The compliance is not the same at each temperature. Was the compliance measured at each temperature of the tests done? Which were the values? Even with the knowledge of the real stifness of the equipment, remais one problem: the deformation of the samples is not uniform at all the sample. In fact, the deformation of the end (extremities) of the samples is not the same at the center zone.To accurate measure this, one high speed 3d video (or laser) system must be used (and this is not easy to do).
6. Page 4. Results. How many samples were used for each kind of test (repetitions)?
7. Page 5. Figure 3. How many samples were used for each kind of test (repetitions)? 1 sample for each kind is poor.
8. Page 7. Line 208. "shown in Figure 6 shows" - too many "shown".
9. Page 8. Equation 4. Variables not defined.
10. Page 9. Figure 7 and Figure 8. The modified model is not good - the predicted stress is very different from the measured values.
11. Page 9. Line 262. "we are unable to offer any explanation" - this sentence is weird. I think the objective of this paper is to offer a model who works, isn't?
12. Page 10. Conclusion 4. ...but the stress values obtained by the model do not agree.
Author Response
Dear Reviewer 3
We would like to thank the reviewer for constructive evaluation and valuable comments. It helped greatly in revising the manuscript. Our response to the reviewer’s comments is given below.
The line numbers specified in the reply are applicable with the track changes mode ‘ON’ of the manuscript.
Improvement in Research design and Methods:
We have taken in the input from the reviewers in the need for improvement in Research design and Methods section. We changed Figure 1 in the experimental part (Page 3: line 110) and also provided an additional reference for clarification (Reference 18) (Page 3: line 106). We have rewritten the methods section on page 9 (line 230-237,255-260,273-284,292-295) for better clarity and have given more detailed information.
Comment 1:
Page 1. Abstract. Line 15. Instead of "experimental testing is post-", better is "experimental testing was post-".
The recommended change is incorporated in the manuscript (page 1: line 16).
Comment 2: Page 2. Line 84. Instead of "Diameter 4 specimens", better is "4 mm diameter specimens".
The recommended change has been incorporated in the manuscript (page 2: line 88-91).
Comment 3: Page 2. Line 92. "heating rate was not expected to lead to microstructural changes". For so high temperatures (700 ºC), I don’t think so.
A reference is added where it is shown, for the steel used in this study, there is no change in the microstructure for temperatures up to 700 ºC (Page 3: line 106).
Comment4: Page 3. Figure 1. The Figure is not well understandable. It is important to also have one schematic drawing.
With the input from the reviewer, we have added a schematic drawing (Page 3: line 106)
Comment 5 Page 3. Line 109. "calculate the machine’s compliance". How was the compliance (or stifness) measured? Compliance is not the same at each temperature. Was the compliance measured at each temperature of the tests done? Which were the values? Even with the knowledge of the real stiffness of the equipment, remains one problem: the deformation of the samples is not uniform at all the sample. In fact, the deformation of the end (extremities) of the samples is not the same at the center zone. To accurate measure this, one high speed 3d video (or laser) system must be used (and this is not easy to do).
Thank you for mentioning the error committed by us. We mean the elastic component of the load – displacement curve. We wrongly used the term machine compliance and we have corrected the same (Page 3: line 122-125)
Comment 6: Page 4. Results. How many samples were used for each kind of test (repetitions)?
2 specimens per test condition where used. In case of considerable variation was observed between the 2 specimens, the 3rd specimen is used. (Page 3: Line 107-108)
Comment 7: Page 5. Figure 3. How many samples were used for each kind of test (repetitions)? 1 sample for each kind is poor.
2 specimens per test condition where used. In case of considerable variation was observed between the 2 specimens, the 3rd specimen is used. (Page 3: Line 107-108)
Comment 8: Page 7. Line 208. "shown in Figure 6 shows" - too many "shown".
The recommended change has been incorporated in the manuscript. (Page 8: Line 230)
Comment 9: Page 8. Equation 4. Variables not defined.
The variables have been defined and incorporated in the manuscript. (Page 8: Line 257 -258,261-262)
Comment 10: Page 9. Figure 7 and Figure 8. The modified model is not good - the predicted stress is very different from the measured values.
We have modified the text according to the above comment (Page 8: Line 297 -299).
Comment 11: Page 9. Line 262. "we are unable to offer any explanation" - this sentence is weird. I think the objective of this paper is to offer a model who works, isn't?
We have deleted the sentence (Page 9: Line 300-301).
Comment 12: Page 10. Conclusion 4. ...but the stress values obtained by the model do not agree.
We have modified Conclusion 4 according to the above comment (Page 11: Line 346).
We would like to thank the reviewer again for a very constructive review and suggestions for reviewing the work.
Thanking you very much in advance.
Yours sincerely
Ashwin Devotta
Sandviken, Sweden

Round 2
Reviewer 1 Report
Review for metals-484756 (second revision)
A modified Johnson-Cook Model for Ferritic Pearlitic Steel in Dynamic Strain Aging Regime
The authors really improved the paper, but the following aspect should be clarified
· For T = 23 ºC the UTS reaches a value about 900 MPa, but from 200 ºC to 400 ºC the UTS is increased. Authors recognized this aspect (lines 134-135 “One zone is between 200 °C and 400 °C where an anomalous behavior is seen with respect to the variation of flow stress with temperature”) but, in my opinion, they should try to include a better explanation of this aspect (increasing of UTS) in the manuscript. This is really interesting, and it should be emphasized as a contribution of the paper.
· Regarding Fig. 4, in zone “dynamic strain aging”, please include in the manuscript an explanation of the different behaviour in case of a strain rate of 60 s-1 in comparison with the other strain rates.
· Please justify why a true strain of 0.1 was selected to develop this research study.
· Please, write the references according to the MDPI format.
· Although the number and the selection of references is adequate, it would be advisable to include some more papers from the journals of MDPI editorial (Materials, Metals, Applied Sciences, etc.) related to the topic of the manuscript.
Author Response
Dear Reviewer 1,
We thank the reviewer for the positive feedback, constructive evaluation and valuable comments. Our response to the reviewer’s comments is given below.
The line numbers specified in the reply are applicable with the track changes mode ‘ON’ of the manuscript.
Anomalous behavior & UTS
In this work, we have used compression testing for studying the flow stress. UTS would be better applicable for flow stress obtained using tensile testing. Instead, we have used varying strain hardening behavior to identify this aspect instead. We recognize the increase in flow stress with increasing strain at temperatures 200ºC - 400ºC which we have mentioned as anomalous behaviour. In the following section with the input from the reviewer, we have identified the anomalous behaviour to be due to dynamic strain aging. We have added the following sentence (page 7: line 202) to relate the anomalous behaviour with dynamic strain aging.
“The increase in flow stress in the temperature zone 200°C-400°C and the negative strain rate sensitivity has been observed as an anomalous behavior in the experimental results. This anomalous behavior is attributed to dynamic strain aging phenomenon.”
Dynamic strain aging at strain rate 60 s-1
With input from the reviewer, we have added the following to the manuscript in Page 4 line 145
“Specifically, at a strain rate of 60 s-1, we observe a significant reduction of the flow stress at 300ºC compared to strain rates 1 s-1 and 5 s-1.”
True strain of 0.1
We would like to thank the reviewer for bringing to focus on this aspect. We have added the following sentence for better clarification (Page 4: Line 150).
“The flow stress at a strain of 0.1 is chosen to avoid the transient conditions present during the early stages of loading”
Reference according to MDPI format and additional references
With the input from the reviewer, we have added relevant and recent references from MDPI Metals and Materials in the area of Johnson-Cook material modeling (References 4 - 7) and modified johnson cook model (References 14-15).
We would like to thank the reviewer again for a very constructive review and suggestions for reviewing the work.
Thanking you very much in advance.
Yours sincerely
Ashwin Devotta
Sandviken, Sweden

Reviewer 3 Report
The manuscript is now ok.
Author Response
Dear Reviewer 3
We thank the reviewer again for constructive evaluation and valuable comments. It helped in revising the manuscript.
English revision
In line with the We have modified English sentences and also checked for grammar/spelling. We have modified (Line 41, 57, 93, 111, 136, 203, 206, 211, 212, 235, 255, 257, 268, 276) to improve readability.
We are thanking you very much in advance.
Yours sincerely
Ashwin Devotta
Sandviken, Sweden

Round 3
Reviewer 1 Report
Only one comment (in the attached file).

Author Response
Dear Reviewer 1,
We would thank you for the input regarding dynamic strain aging at varying strain rates (1s-1 & 5s-1) and 60 s-1.
With the reviewer’s input, we have added P1 (page 4 line 143-146), P2 (page 5 line 159-162) and P3 (Page 7 line 221-227).
P1: With strain rate increase from 1 s-1 to 5 s-1 and 60 s-1 in the programmed temperature zone of 200 °C- 400 °C, reduced flow stress indicates negative strain rate sensitivity.
P2: At strain rates 1 s-1 & 5 s-1, the flow stress increases in the temperature zone 200 °C- 400 °C. On the other hand, at the strain rate of 60 s-1, the flow stress decreases marginally from 200 °C- 300 °C and slightly increases from 300 °C- 400 °C.
P3: At 1 s-1 and 5 s-1, the continuous flow stress increase in the temperature zone (200 °C- 400 °C) and the observed serrations show a strong presence of DSA. Whereas at 60 s-1, the absence of serrated flow and a marginal flow stress increase from 300 °C- 400 °C shows a relatively weaker presence of DSA. This clearly shows that the DSA regime is strongly influenced by the strain rate – temperature range. To capture DSA active in the above strain rate - temperature range, the JC parameters A and B are to be modeled as a function of temperature and strain rate.
We would like to thank you again for your valuable input.
Yours sincerely
Ashwin Devotta
Sandviken, Sweden
